# Impact of LS Mutation on Pharmacokinetics of Preventive HIV Broadly Neutralizing Monoclonal Antibodies: A Cross-Protocol Analysis of 16 Clinical Trials in People without HIV

**DOI:** 10.3390/pharmaceutics16050594

**Published:** 2024-04-27

**Authors:** Bryan T. Mayer, Lily Zhang, Allan C. deCamp, Chenchen Yu, Alicia Sato, Heather Angier, Kelly E. Seaton, Nicole Yates, Julie E. Ledgerwood, Kenneth Mayer, Marina Caskey, Michel Nussenzweig, Kathryn Stephenson, Boris Julg, Dan H. Barouch, Magdalena E. Sobieszczyk, Srilatha Edupuganti, Colleen F. Kelley, M. Juliana McElrath, Huub C. Gelderblom, Michael Pensiero, Adrian McDermott, Lucio Gama, Richard A. Koup, Peter B. Gilbert, Myron S. Cohen, Lawrence Corey, Ollivier Hyrien, Georgia D. Tomaras, Yunda Huang

**Affiliations:** 1Vaccine and Infectious Disease Division, Fred Hutchinson Cancer Center, Seattle, WA 98109, USA; yzhang2@scharp.org (L.Z.); adecamp@scharp.org (A.C.d.); cyu@scharp.org (C.Y.); asato@scharp.org (A.S.); hangier@fredhutch.org (H.A.); jmcelrat@fredhutch.org (M.J.M.); hgelderb@fredhutch.org (H.C.G.); pgilbert@fredhutch.org (P.B.G.); lcorey@fredhutch.org (L.C.); ohyrien@fredhutch.org (O.H.); 2Duke University Medical Center, Durham, NC 27705, USA; kelly.seaton@duke.edu (K.E.S.); nicole.yates@duke.edu (N.Y.); georgia.tomaras@duke.edu (G.D.T.); 3Vaccine Research Center, National Institute of Allergy and Infectious Diseases, Bethesda, MD 20892, USAmpensiero@niaid.nih.gov (M.P.); adrian.mcdermott@sanofi.com (A.M.); lucio.gama@nih.gov (L.G.); richard.koup@nih.gov (R.A.K.); 4The Fenway Institute, Boston, MA 02215, USA; kmayer@fenwayhealth.org; 5Laboratory of Molecular Immunology, The Rockefeller University, New York, NY 10065, USA; mcaskey@rockefeller.edu (M.C.); nussen@mail.rockefeller.edu (M.N.); 6Ragon Institute of Mass General, MIT and Harvard, Cambridge, MA 02139, USA; kstephen@bidmc.harvard.edu (K.S.); bjulg@mgh.harvard.edu (B.J.); 7Center for Virology and Vaccine Research, Beth Israel Deaconess Medical Center, Boston, MA 02215, USA; dbarouch@bidmc.harvard.edu; 8Columbia University Irving Medical Center, New York, NY 10032, USA; mes52@cumc.columbia.edu; 9Department of Medicine, Division of Infectious Diseases, Emory University School of Medicine, Atlanta, GA 30322, USA; sedupug@emory.edu (S.E.); colleen.kelley@emory.edu (C.F.K.); 10Department of Biostatistics, University of Washington, Seattle, WA 98195, USA; 11Institute for Global Health and Infectious Diseases, University of North Carolina at Chapel Hill, Chapel Hill, NC 27514, USA; mscohen@med.unc.edu; 12Departments of Medicine and Laboratory Medicine, University of Washington, Seattle, WA 98195, USA; 13Department of Global Health, University of Washington, Seattle, WA 98195, USA

**Keywords:** monoclonal antibodies, LS mutation, two-compartment model, population pharmacokinetics modeling, HIV prevention, targeted maximum likelihood estimation (TMLE)

## Abstract

Monoclonal antibodies are commonly engineered with an introduction of Met428Leu and Asn434Ser, known as the LS mutation, in the fragment crystallizable region to improve pharmacokinetic profiles. The LS mutation delays antibody clearance by enhancing binding affinity to the neonatal fragment crystallizable receptor found on endothelial cells. To characterize the LS mutation for monoclonal antibodies targeting HIV, we compared pharmacokinetic parameters between parental versus LS variants for five pairs of anti-HIV immunoglobin G1 monoclonal antibodies (VRC01/LS/VRC07-523LS, 3BNC117/LS, PGDM1400/LS PGT121/LS, 10-1074/LS), analyzing data from 16 clinical trials of 583 participants without HIV. We described serum concentrations of these monoclonal antibodies following intravenous or subcutaneous administration by an open two-compartment disposition, with first-order elimination from the central compartment using non-linear mixed effects pharmacokinetic models. We compared estimated pharmacokinetic parameters using the targeted maximum likelihood estimation method, accounting for participant differences. We observed lower clearance rate, central volume, and peripheral volume of distribution for all LS variants compared to parental monoclonal antibodies. LS monoclonal antibodies showed several improvements in pharmacokinetic parameters, including increases in the elimination half-life by 2.7- to 4.1-fold, the dose-normalized area-under-the-curve by 4.1- to 9.5-fold, and the predicted concentration at 4 weeks post-administration by 3.4- to 7.6-fold. Results suggest a favorable pharmacokinetic profile of LS variants regardless of HIV epitope specificity. Insights support lower dosages and/or less frequent dosing of LS variants to achieve similar levels of antibody exposure in future clinical applications.

## 1. Introduction

Broadly neutralizing monoclonal antibodies (mAbs) are being considered as a potential strategy to prevent and treat human immunodeficiency virus (HIV) disease [1,2,3,4,5,6]. Particularly, the recent Antibody Mediated Prevention (AMP) studies of VRC01, an HIV broadly neutralizing mAb intravenously administered every 8 weeks, demonstrated no overall protection but a prevention efficacy of 75% against VRC01-neutralization-sensitive HIV viruses [7,8]. The AMP studies also established that prevention efficacy was related to the predicted neutralization ID80 titer (PT80) of VRC01 as an integrated biomarker of VRC01 serum concentration at the time of HIV acquisition, as well as the in vitro VRC01 neutralization sensitivity against the acquired virus [7,8,9]. These findings suggest that greater mAb-mediated prevention efficacy could be achieved by improving the neutralization activity of a mAb against circulating viruses, and/or by prolonging elevated concentrations of the mAb over time [10]. The next generation of HIV mAbs showed promising improvements compared to VRC01 for prevention, therapy, and potential for long-term virologic control or remission; in vitro analyses of a new series of mAbs highlight more target options for binding sites on the HIV trimer and generally improved neutralization potential [11,12,13]. To improve the pharmacokinetic (PK) profiles, the most common approach has been the use of the LS mutation (M428L/N434S) in the constant fragment crystallizable (Fc) region of immunoglobin G (IgG)-based mAbs [14,15,16], which could potentially also improve in vivo activity [17]. 

The LS mutation enhances the Fc region’s binding affinity for the neonatal Fc receptor (FcRn) at a low pH (e.g., lysosomal compartments and vaginal lumen) without affecting its release at pH 7.4 [14,15,16]. The FcRn is expressed in endothelial cells of the vasculature and some myeloid cells, where it can rescue IgG from lysosomal degradation by binding these molecules within this low pH compartment and then releasing them back into circulation [15,18]. Importantly, the tissue distribution of FcRn enables the delivery of IgG mAbs to relevant mucosal compartments and may thereby contribute to HIV protection at the site of exposure [14]. Increased persistence of anti-HIV neutralizing antibodies in mucosal tissue associated with the LS mutation could have significant advantages in preventing HIV acquisition, as increased maintenance at the portals of entry might allow for consistent protective levels with less frequent administrations and/or a reduction in dosing.

So far, clinical studies of several anti-HIV LS variant mAbs demonstrated extended durability as measured by improved elimination half-life [19,20]. However, less is known about how the LS modification could improve other aspects of PK profile, including clearance rate, distributional volumes, or average concentrations over a dosing interval. In addition, there have not been systematic PK analyses of available HIV mAbs to understand whether the benefits of the LS modification are consistent for mAbs targeting different epitope sites on the HIV trimer.

Here, we hypothesized that across multiple mAbs the LS modification would significantly and predictably improve PK profiles over the parental mAb as determined by measures of extended elimination rates and reduced clearance. To that end, we analyzed data from 16 clinical trials of 11 HIV-1 mAbs with and without the LS mutation. Using these data, we developed population PK models for each of the 11 HIV-1 mAbs and investigated differences in PK parameters between five pairs of parental and LS mutation versions. We used the targeted maximum likelihood estimation (TMLE) [21] method to examine how key PK parameters changed between parental and LS variants. TMLE allows for rigorous adjustment for demographic and clinical factors that affect PK using an ensemble of learners, estimated under the counterfactual scenario that the two versions of mAbs (parental and LS) were assessed in the same study population. In addition, we accounted for variabilities and correlations of PK parameter estimates in assessing the statistical precision of the comparisons. Overall, we found that the LS modification reliably improved the PK profiles for each assessed anti-HIV mAb pair in a largely consistent manner.

## 2. Materials and Methods

### 2.1. Data

We analyzed PK data for 5 parental HIV-1 mAbs targeting three different epitopes on the HIV trimer: VRC01 (CD4 binding site), 3BNC117 (CD4 binding site), PGDM1400 (V1V2 glycan), PGT121 (V3 glycan), 10-1074 (V3 glycan), and their LS variants (Appendix A). We also analyzed data from VRC07-523LS, an engineered LS variant derived from VRC01 [22]. We included individual-level data collected from 16 phase 1 clinical trials of these mAbs [19,20,23,24,25,26,27,28,29], conducted in compliance with all relevant ethical requirements, including obtaining participant consent. All trials aimed to evaluate the given study product(s), administered via subcutaneous (SC) or intravenous (IV) route, in healthy adult participants without HIV. Details of the clinical trials and assays used to generate the PK data can be found in Table 1.

### 2.2. PK Assays

#### 2.2.1. Enzyme-Linked Immunosorbent Assay (ELISA)

ELISA was used to quantify the mAb concentrations in human serum for VRC01 [25,26], VRC01LS [19], VRC07-523LS [20,31], 3BNC117 [23,34], 3BNC117LS [27], 10-1074 [37], and 10-1074LS [35] in 10 of the 16 clinical trials. Briefly, for a series of diluted samples, serum antibody levels were measured using optical density readings based on antibody-specific anti-idiotype binding. Serum concentrations were then determined from a standard curve using a 5PL curve fit. For VRC01 [25,26], VRC01LS [19], and VRC07-523LS [20,31,32], the specific anti-idiotype antibodies were developed and purified by the Vaccine Research Center (VRC), National Institute of Allergy and Infectious Diseases (NIAID) at the National Institutes of Health (NIH). For 3BNC117 [23,34], 10-1074 [37], and their LS variants [27,35], the specific anti-idiotype antibodies were developed and purified by the Duke Vaccine Immunogenicity Program [23]. For this assay, the lower limits of quantification (LLoQ) varied over the range of 0.12–1.1 mcg/mL depending on the specific mAb and study (Appendix A).

#### 2.2.2. Singulex Assay

The Singulex assay was used to quantify concentrations in human serum for VRC01/VRC01LS [30] in the HIV Vaccine Trials Network (HVTN) 116 phase 1 clinical trial. Briefly, concentrations in the (diluted) samples were measured using Singulex paramagnetic microparticle (MP) beads using the Singulex Erenna^®^ Capture Antibody Labeling Kit (Singulex/EMD Millipore) according to the manufacturer’s protocol. The Erenna detects single molecules of VRC01/VRC01LS as photons over a read time. Each measurement of photons above a threshold was considered a detected event and the sum of all photons from these events over the read time (i.e., signal intensity) is called the event photon (EP) measurement for the sample or standard. The EP curve of the VRC01 standard was used to calibrate the VRC01/VRC01LS concentrations in serum samples. The median LLoQ across runs was 2.31 × 10^−4^ mcg/mL.

#### 2.2.3. PK Binding Antibody Multiplex Assay (BAMA)

The BAMA assay derived on a Bio-Plex instrument was used to quantify concentrations in human serum for VRC07-523LS [20,28], PGDM1400 [24,36], and PGT121 [29,33,38] under the oversight of the Quality Assurance for Duke Vaccine Immunogenicity Programs. This anti-idiotype (anti-ID) assay measured concentrations by its ability to bind anti-idiotype antibody captured on fluorescent magnetic beads. The Bio-Plex software (https://www.bio-rad.com/en-us/product/bio-plex-data-pro-software?ID=LSVAC015, accessed on 26 March 2024) provides 2 readouts: a background-subtracted median fluorescent intensity (MFI), where background refers to a plate level control (i.e., a blank well containing antigen-conjugated beads run on each plate), and a concentration based on a standard curve using a 5PL curve fit. For this assay, the LLoQ varied over the range of 0.023–0.50 mcg/mL depending on the specific mAb and study (Appendix A).

### 2.3. Population PK Modeling

We modeled serum mAb concentrations over time using non-linear mixed effects modeling with the Monolix software system (Version 2021 R2) (https://lixoft.com/products/monolix/, accessed on 26 March 2024). We estimated population parameters, including the population mean and variance of each PK parameter and parameters of the error term variance by fitting the non-linear mixed effects models using the method of maximum likelihood implemented with the Stochastic Approximation Expectation Maximization (SAEM) algorithm as implemented in Monolix. We described PK following intravenous (IV) or subcutaneous (SC) mAb administration by an open 2-compartment disposition model with first-order elimination from the central compartment. A 2-compartment model assumes that antibody clearance is bi-phasic, which has been routinely used to describe the PK of IgG mAbs [39]. The model was parameterized in terms of the central volume (Vc, L), clearance from the central compartment (CL, L/day), peripheral volume (Vp, L), and intercompartmental clearance (Q, L/day). For SC administrations, a depot compartment was added to characterize first-order absorption via the SC route with 2 parameters: bioavailability (F, %) and absorption rate (ka, day^−1^).

### 2.4. Error Model and Random Effects Specification

For each population PK model, we considered population-level (average) parameters as fixed effects and inter-participant variation of these parameters as random effects with standard deviations (ω). We did not include random effects for absorption rate and bioavailability due to limited data availability to estimate these parameters across all mAbs. We fitted PK models to serum concentrations, assuming normally distributed error terms. Concentrations below the LLoQ were treated as left-censored (Appendix A). We assessed two structures for the conditional variance of the error term, given the random effects, assuming the following: (1) the error was proportional to the conditional expectation of the concentration; or (2) the residual variance was the sum of 2 terms, one constant term and another term proportional to the conditional expectation of the concentration. For each mAb, we selected the final error model structure based on the model with the lowest corrected Bayesian Information Criteria (BIC) [40]. In addition, we assumed the variance–covariance matrix of the random effects to be diagonal with heterogeneous variances for Vc, CL, Vp, and Q. Individual-level PK parameters were assumed to follow log-normal distributions. For PGDM1400LS, there were limited data to estimate the distribution phase and hence the random effects were not included for Vp and Q.

We predicted participant-specific model parameters using their empirical Bayes estimates (EBEs), defined as the most probable value given the estimated population parameters and the data from the given participant. We predicted these (random) individual-specific parameters by the mode of their conditional posterior distribution. They were subsequently used to predict the most probable trajectory of the serum concentration for the corresponding participant.

In addition to the model parameters (CL, Vc, Vp, and Q), we derived four individual-level PK features from the base population PK model without covariate adjustment: distribution half-life, elimination half-life, dose-normalized area under the curve (AUC), and predicted concentrations at 4 weeks post-infusion following a single IV bolus administration of 1.4 g of given mAb. A fixed dose of 1.4 g was considered to approximate a body-weight-based dose of 20 mg/kg for a 70 kg individual. The dose-normalized AUC values and predicted concentrations were computed for each individual participant using their individual-level parameters (i.e., EBEs) as predicted from the fitted PK model for the given mAb.

We used the Lin’s concordance correlation coefficient (CCC) [41] to assess the concordance between individual-level parameter estimates via fitting a model to each trial separately vs. a model fitted to the pooled trials for each mAb. The CCC is a composite measure of precision (Pearson’s correlation coefficient) and accuracy (bias). The quality of concordance is interpreted as poor when CCC < 0.9, moderate when 0.9 < CCC< 0.95, and substantial when CCC > 0.95 [42].

### 2.5. TMLE Methods

Although all 16 clinical trials were carried out in similar study populations supporting overlapping distributions of baseline characteristics between the comparison groups, the parental vs. LS variant groups were not randomized. Therefore, to formally compare PK features between the parental and LS variants, we used the TMLE method [21]. TMLE is an alternative to standard linear or non-linear regression that can improve robustness and efficiency and reduce confounding bias, especially for comparisons between non-randomized groups. Using this method, for each mAb pair we estimated the mean of each PK feature for each mAb in the pair, and the average treatment effect (ATE) of LS vs. parental of the pair, adjusted for potential confounders and predictors of PK variability: baseline body weight (kg), creatinine clearance (mL/min), age (years), and sex at birth (female vs. male), as implemented via the *tmle* package (version 1.5.0.2) in R (version 4.2.1) [43]. Notably, because it was challenging to precisely and accurately estimate these two features without richly sampled kinetic data proximal to product administration, the Q parameter and the distribution half-life were not compared between the parental and LS variants. A sensitivity analysis was also performed to account for single vs. co-administration for parental/LS mAb pairs. All TMLE estimation results of means and ATEs were averaged over 20 runs with a fixed random seed on top of the 10-fold cross-validation estimation procedure to ensure stability of the estimates. The set of learning algorithms used by TMLE for estimating the mean PK feature and ATE conditional on baseline covariates included the following: SL.glm, SL.step, SL.ranger, SL.earth, SL.glmnet, and SL.mean [44]. In addition, to account for variability and co-variability of the individual-level estimates for each PK feature, due to the fact that they were derived from a common population PK model, a bootstrap procedure based on 500 sampled-with-replacement datasets was used to estimate the variance of the estimates for mAb (parental or LS), derive the 95% confidence intervals (CIs), and test for a non-zero mean difference between the two mAbs in each parental/LS pair. We used the Holm procedure [45] to adjust the resulting p-values for comparisons of the multiple PK features within a given mAb pair.

## 3. Results

### 3.1. Study Population Characteristics

Here we analyzed the PK properties of anti-HIV mAbs targeting three different epitopes on the HIV-1 trimer: CD4 binding site (VRC01, 3BNC117), V3 glycan (PGT121, 10-1074), and V1V2 glycan (PGDM1400) (Table 1 and Appendix A). For each mAb, we evaluated both the parental formulation and its LS-formulated variant. For VRC01, we also compared the PK features with VRC07-523LS, the LS variant of another CD4-binding-site-targeting mAb that was engineered by the same developer and is a clonal relative of VRC01 from the same donor [22].

We leveraged data from a total of 16 phase 1 clinical trials, with 95% of the 583 participants enrolled from clinical sites in the US (Table 1). In terms of the study populations, the 10-1074 and 3BNC117 studies included participants who were slightly older (medians 41–49 years), weighed more (medians 79–83 kgs), and comprised fewer females (20–46%). For the remaining mAbs, participants’ age ranged over medians of 26–31 years, weights ranged over medians of 71–76 kgs, and the proportion of female participants ranged from 48–67% (Appendix A).

### 3.2. Population PK Model Fitting

Serum concentrations of all the mAbs exhibited bi-phasic clearance following administration (Appendix A). For each mAb variant assessed in multiple clinical trials, we fitted models both separately by trial and then with a single model combining the data across trials to determine the suitability of pooling data across studies for a given mAb. For PK features, except Vc and Vp, there was at least moderate concordance (CCC > 0.9) between the trial-level and combined models. For Vc, the estimates in the SC population tended to be more deviant between the trial-level and combined models (triangles in the Vc row of Appendix A). The Vc and bioavailability (F) parameters were mathematically non-identifiable in SC groups without precise estimation of Vc from groups receiving IV administration, and thus pooling the data across studies could improve the accuracy and precision of estimates of Vc in the SC groups by increasing the sample size for the IV data. Subsequently, we used the combined model for the remaining analyses.

### 3.3. Covariate-Unadjusted Base PK Models for Each mAb

The final, covariate-unadjusted 2-compartment models generally exhibited good fit to all the data. Goodness of fit at the individual and population levels was demonstrated for all mAbs based on analysis of the residuals and model-predicted concentrations (Appendix A). The population PK parameters for all the mAbs are depicted in Appendix A. Population-level bioavailability after SC administration compared to IV estimates ranged from 39–78%. Figure 1 and Figure 2 show the distributions of the PK parameter estimates obtained from the base population PK model without covariate adjustment, which are the basis for the subsequent comparisons. The unadjusted individual parameters for Vc and CL were consistent across the parental and LS variants across the mAbs. The CL was notably slower for LS variants compared to parental variants, an expected trend for the LS mutation. The Vc tended to be slightly lower for the LS variants as well, but not uniformly, potentially indicating a smaller distribution space (i.e., higher maximum concentration) in the central compartment for LS variants. The Q and Vp parameters showed notable variation across the mAbs (Figure 1). Variation in these parameters may be attributable to varying sampling schemes where first-in-human trials of a mAb would have more frequent early sampling to precisely determine the distribution phase than subsequent trials of a product. For PGDM1400LS, limited data precluded estimation of the inter-individual variation for Vp and Q, and therefore all individual-level predicted parameters were equivalent to the population estimate. Overall, there was a general pattern that the LS variant had both lower Vp and Q (except for VRC01), suggesting a different distribution process to the peripheral compartment with the LS mutation. In addition, the LS mutation extended elimination half-lives in all mAb pairs. Consistent with the improved elimination half-life and differences in central compartment distribution, drug exposure also appeared much higher for the LS variants as determined by dose-adjusted AUC and predicted week 4 concentration with 1.4 g IV dose (Figure 2).

### 3.4. Covariate-Adjusted Differences in LS and Parental PK

We next used a combination of bootstrapping and TMLE to estimate the PK parameters and features for each mAb pair (parental vs. LS) after adjusting for baseline covariates. The individual-level PK parameter estimates obtained using TMLE depict the counterfactual value of these parameters/features, providing an unbiased comparison of these values as if the same group of individuals received the parental and LS mAbs (Figure 3 and Figure 4). Likely due to the largely comparable characteristics between participants who received the parental vs. LS mAbs (Appendix A), the direction of the LS benefit in these TMLE-adjusted estimates, as well as the average estimates, was consistent with those observed in the unadjusted estimates as shown in Figure 2.

Specifically, we found significant differences in CL, Vc, and Vp between parental and LS variants for each mAb pair, except for Vc and Vp for PGDM1400/LS (Figure 5, Appendix A). Among the significant differences, the changes in parameters were all in the same direction, to the left of 1.0 or to the right of 1.0 in terms of the LS/parental ratio, as shown in the forest plot of Figure 5. Specifically, the mean CL decreased in a range of 4.2- to 9.1-fold (*p* < 0.001 for all), the mean Vc decreased in a range of 1.4- to 1.7-fold (*p* ≤ 0.03, except PDGM1400, *p* = 0.157), and the mean Vp decreased in a range of 1.7- to 3.3-fold (*p* < 0.001 all except PGDM1400 not tested).

In addition, the direction of the changes for the PK parameters was associated with both extended elimination clearance and higher observed drug exposure for the LS variant compared to the parental for each mAb pair, all statistically significant at *p* < 0.001 (Figure 5 and Appendix A). The elimination half-life for the LS variant was extended by a mean range of 2.7- to 4.1-fold, the dose-normalized AUC was increased by a mean range of 4.1- to 9.5-fold, and the predicted concentration at 4 weeks post-administration was increased by a mean range of 3.4- to 7.6-fold.

To illustrate the overall impact of the LS mutation on the PK profiles, we simulated PK curves for each mAb administered intravenously at 1.4 g using the mean TMLE-estimated parameters (Figure 6). A general visual pattern emerged comparing the parental to the LS variant: the LS variant concentration was initially higher (i.e., higher peak serum concentration), then all mAbs had a sharp drop in serum concentration during the distribution phase that was less pronounced with the LS mutation. Finally, the LS variant remained at higher concentration with slower decay throughout the elimination phase. A slight exception to this pattern was observed for the PGDM1400 pair, wherein the parental and LS had similar values immediately following administration. In addition, PDGM1400LS exhibited no prominent distribution phase, likely due to the limited available data for this LS variant. Consequently, for PGDM1400LS there appeared to be one single decay rate over time, and it remained at higher serum concentrations than the parental variant throughout.

### 3.5. Covariate-Adjusted Differences in VRC01 vs. VRC07-523LS PK

Lastly, we examined the differences between VRC01 and VRC07-523LS to gain more insights into when a mAb has additional modifications other than the LS mutation. While VRC01 is not the parental formulation of VRC07-523LS—the parental VRC07 has never been tested in a clinical setting—VRC07 was engineered from VRC01. Therefore, PK differences between VRC01 and VRC07-523LS cannot be solely attributed to the LS mutation. The covariate-unadjusted individual parameter estimates for VRC07-523LS were generally different from VRC01 (Figure 1). Compared to VRC01, VRC07-523LS also exhibited a longer elimination half-life, a higher does-normalized AUC, and a higher predicted concentration 4 weeks post-administration (Figure 2). Using TMLE, we found significant differences for all parameters except Vp between VRC01 and VRC07-523LS (Figure 3 and Figure 4 and Appendix A). The mean CL decreased 2.9-fold (*p* < 0.001) between VRC01 and VRC07-523LS, a smaller decrease than that observed among the five pairs of parental/LS mAbs. The mean central volume for VRC07-523LS increased 1.6-fold (*p* < 0.001) compared to VRC01, which is the opposite direction of the trend observed comparing LS to parental variants where the volumes decreased. No significant difference was detected in Vp (*p* = 0.247 d). The elimination half-life for VRC07-523LS was extended over VRC01 by 3.1-fold (*p* < 0.001), similar to the improvement observed with LS mutation for the parental/LS variant mAb pairs. The drug exposure was also higher for VRC07-523LS compared to VRC01, with a 3.0-fold increase (*p* < 0.001) in dose-normalized AUC and a 2.7-fold increase (*p* < 0.001) in serum concentration at 4 weeks post-administration, but these increases were below the ranges observed in the five pairs of parental/LS mAbs. Overall, the PK changes of VRC07-523LS compared to VRC01 appeared somewhat different from what was observed by directly comparing parental to LS among the other mAbs, including VRC01LS vs. VRC01.

## 4. Discussion

To our knowledge, this is the first unified pharmacokinetics analysis of multiple anti-HIV mAbs from multiple clinical trials to examine the effect of LS mutation on their PK properties upon IV or SC administration in people without HIV. In this cross-protocol analysis of mAb serum concentrations over time, we leveraged data for five pairs of parental vs. LS variant anti-HIV mAbs evaluated in 16 different phase I clinical trials involving a total of 583 participants from the US, Europe, and Africa. Importantly, our comparison analyses adjusted for participants’ clinical and demographic characteristics across independent clinical trials. Overall, we found consistent and relatively predictable PK improvement associated with the LS modification compared to the parental mAbs. These findings are important because no other meta-analyses of anti-HIV mAbs have been previously performed to understand systematically the effect of LS mutation; the learned knowledge could be applied to the research and development of LS mAbs, and in the design of clinical trials evaluating LS mAbs, given the knowledge of their parental formulation.

Specifically, we observed bi-phasic clearance of serum concentrations across all pairs of parental/LS mAbs, supporting the use of an open two-compartment disposition model with first-order elimination as the best fitting PK model to describe these data, in agreement with previous analyses for most of these mAbs individually [19,20,23,24,25,26,27,28,29]. In addition, we successfully fit population two-compartment models to each mAb using non-linear mixed effects modeling. The parameter estimates of these anti-HIV parental mAbs from our models were consistent with those reported in a previous meta-analysis of non-HIV parental mAbs [39]; to our knowledge, no meta-analysis of anti-HIV or non-HIV LS mAbs has been reported. For each pair of these parental/LS mAbs, we observed that LS mAbs had a reduced CL rate resulting in a 2.7- to 4.1-fold extended elimination half-life and 4.1- to 9.5-fold higher drug exposure in serum. We also observed that the distributional properties were different in the LS mAbs, generally with 1.4- to 3.3-fold lower central and peripheral volumes (Vc and Vp), also contributing to increased dose-normalized concentrations in serum. These results exemplify the complex biological implications of the LS mutation. It is believed that the LS mutation extends mAb persistence in serum, primarily because the FcRn on various cells helps to reduce degradation and clearance and to improve recycling of the mAb back into circulation. Consequently, as hypothesized, all the LS mAbs here have extended elimination half-lives and reduced CL. Coupled with the observed uniform reductions in the volume parameters with the LS mutation, these results suggest that the general distribution of the mAbs in the serum and tissue was likely altered by FcRn binding. Overall, the increased serum drug exposure and overall improvement to the PK profile seem to result from a combination of both the improved half-life and the distributional differences. Critically, as applied to future clinical designs, these improvements suggest reduced dosing frequency needed for LS variants; for example, a 2-month interval for parental mAbs could be reduced to a 6-month interval for their LS variants.

Importantly, we also found that the LS mutation, when compared to the parental formulation, had a similar effect on PK regardless of the specific epitopes on the HIV trimer that the mAbs target. This could potentially be attributed to the fact that the LS modification is a relatively small change of two amino acids (AAs) in the constant Fc region of a relatively large molecule of approximately one thousand five hundred AAs. The quantitative differences in the parameters were not widely different across the mAbs, generally falling within 0.5 log_10_ changes of each other. It is important to note that these estimated differences between each parental/LS pair adjusted for potential imbalances in participants’ characteristics that might influence PK. The relative consistency of these parental/LS differences across mAbs indicates that prediction of PK for a mAb with an untested LS mutation may be performed with limited uncertainty based on the PK of the parental formulation. This is important knowledge for the development and planning of future trials of newly discovered HIV-1 mAbs—and potentially for mAbs targeting other pathogens—and their LS modifications. Phase 1 studies remain critical to assess safety and PKs, specifically as the manufacturing process could introduce unexpected variables even though the LS mutation is a relatively small change in the context of the size of the antibody. This analysis helps to provide a baseline for what should be expected when assessing PKs for an LS-modified anti-HIV IgG mAb.

Lastly, we found that PK features were largely consistent across trials for the same mAb. This highlights the general conservation of the PK biological properties and the reproducibility of these phase 1 studies. Particularly, the estimates of CL, half-lives, and simulated drug exposures were remarkably concordant whether the trials were modeled in combination or separately for a given mAb. We did, however, observe some variation in the volume parameters. The Vc parameter is meant to approximate serum blood volume (~3L) [46] but may exhibit variation, as it also functions as a scaling parameter between the administered dose mass and the observed serum concentration. Thus, the Vc differences observed between the trials could be attributable to varying sampling schemes or differing assays rather than differences in demographic or biological factors.

There were several limitations to this analysis. First, the included trials had different PK sampling time points because some were first-in-human trials of the mAbs, with more detailed and intensive sampling especially after the single administration, and others were follow-on trials of the same mAb, with different dosing levels, administration routes, or co-administration with other mAbs. PK studies may be particularly sensitive to sampling schemes, specifically for estimation of the distributional phase (i.e., early kinetics). For PGDM1400, this challenge was observed by the difficulty in estimating the inter-subject variation for distribution PK parameters (Q and Vp). Despite this limitation, the differences in mean adjusted estimates of Vp between parental and LS mutations were in a tight range (1.7- to 3.3-fold). 

Second, because the context of this research is for the prevention of HIV acquisition, as opposed to therapeutic treatment, limited sampling time points immediately following the SC administrations were available from the 16 phase 1 trials. Therefore, our ability to discern the effect of the LS mutation on first-order absorption properties (i.e., the absorption rate from the SC space and bioavailability) of the mAbs was limited. To precisely estimate the absorption rate, sampling schemes with finer granularity are required generally within the first week of product administration. However, since serum concentrations are expected to provide the highest protection during this early interval, PK studies in the context of prevention typically do not focus on these early time points as opposed to later time points. Consequently, there were no pairs of parental and LS variants in PK studies that were run with SC groups under a common design to precisely compare this parameter. The limited SC data also prevented precise estimation of bioavailability, as that requires studies with both SC and IV groups for a given LS/parental mAb pair. When we were able to estimate bioavailability, we observed a large range of estimates across the mAbs (i.e., 38% to 76%) that is consistent with a wide range of bioavailability observed generally with mAbs [47]. However, we have limited available data to conclude with certainty whether the range observed here reflects true biological variability across these mAbs, or whether it is a combined result of limited sample size to precisely estimate Vc and F together with different PK sampling time points across the mAb pairs.

Third, the use of different binding antibody assays may have contributed to PK variability in the analysis, particularly when comparing across studies. Since concentration measurements on the same set of sera were only done by a given assay for each mAb, it is infeasible to estimate an assay effect. Therefore, the generalizability of results may be limited when trying to compare parameters or concentrations across the mAbs when different assays were used. 

Fourth, while this analysis illustrated common changes to the PKs of mAbs comparing parental to LS variants, it is limited to conclusions made using serum concentrations. In a recent example using a mAb LS modification to prevent COVID-19 [48] in cynomolgus monkeys, the LS modification increased standardized uptake of sotrovimab but did not change the tissue–blood ratios compared to its parental counterpart. Regarding HIV prevention mAbs, preliminary data showed that VRC01LS and VRC07-523LS have different tissue distribution in SIV/SHIV-free naïve/uninfected macaques, and VRC01 and VRC01LS have different rectal tissue to blood ratios in concentrations [49]. Yet it is unclear how much of this distinction is due to possible non-specific binding (Fab) or unequal interaction with FcRn receptors. Additionally, while the LS mutation predictably increases drug persistence via mutation only to the Fc region and not to the Fab regions, it should always be confirmed that the neutralization potential of the LS-mutated mAb has not been altered compared to the parental version.

Fifth, currently available data were collated from trials generally conducted in the US, with relatively limited data from other regions. Therefore, future updates of this analysis are needed when data of the studied or new anti-HIV mAbs become available from on-going and planned trials conducted in other regions. On the other hand, we do not anticipate a significant regional effect as demonstrated, for example, in the analysis of PK data from the AMP trials that were conducted globally in the Americas, Europe, and South Africa [9,44]. 

Lastly, our analysis was limited to three classes of anti-HIV mAbs and may not translate to mAbs that target other binding sites on the HIV trimer (e.g., 10E8 targeting the MPER). Translation of the presented findings to non-HIV mAbs would also require additional research.

Credited to favorable tolerability, safety, effectiveness, and durability [50], there has been a rapid increase of mAb use for disease treatment and prevention. As of January 2023, the Food and Drug Administration approved more than 165 mAb products for use across a variety of diseases, including rheumatoid arthritis, cancer, Crohn’s disease, psoriasis, and hyperlipidemia [51], and to prevent infections such as respiratory syncytial virus (RSV) [52] and severe acute respiratory syndrome coronavirus 2 (SARS-CoV-2) [53]. As more mAbs are being developed, and increasingly used for prevention and treatment of infections, cancer, and other diseases, our investigation provides important knowledge based on a systematic analysis of five pairs of anti-HIV mAbs for the research and development of LS-formulated mAbs, especially those against HIV. Our findings provide benchmarking for what could be expected from LS mutations on emerging new mAbs regardless of indication. More research is needed to understand whether the impact of the LS modification on PKs holds in mucosal compartments and across mAbs against other non-HIV infections or diseases.

## Figures and Tables

**Figure 1 pharmaceutics-16-00594-f001:**
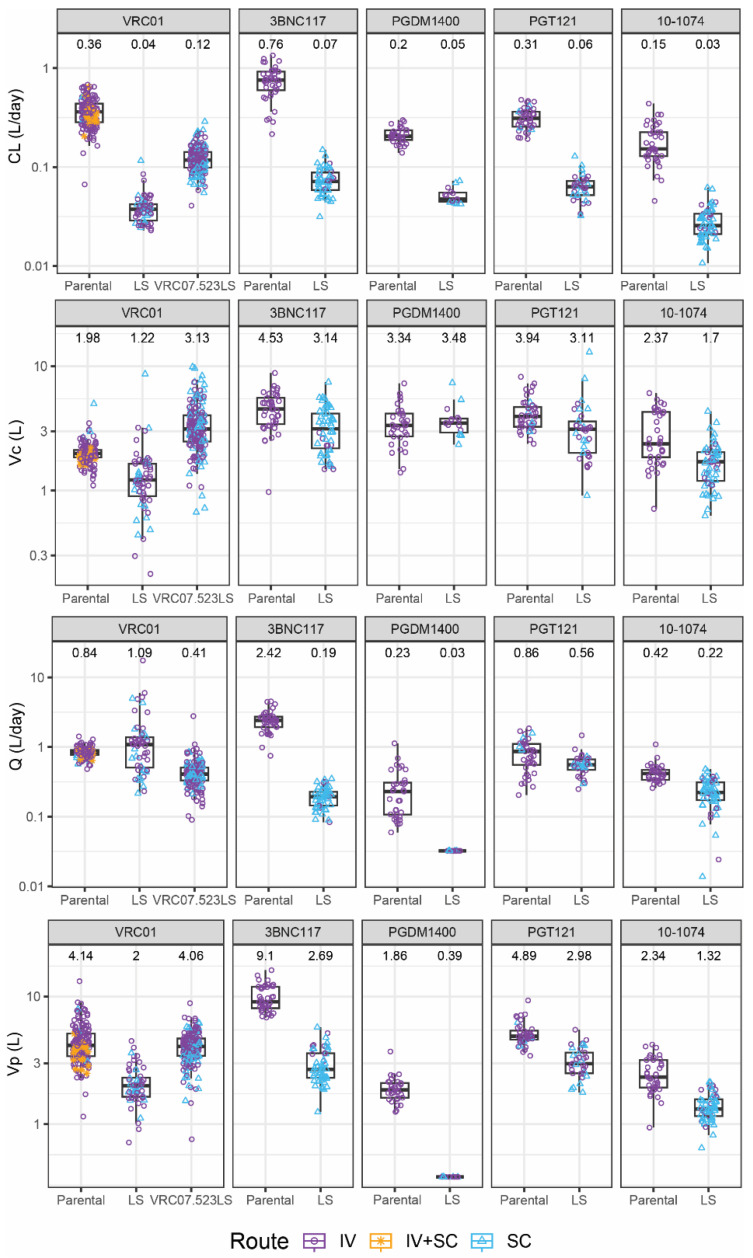
Distributions of population-PK-model-estimated covariate-unadjusted individual-level pharmacokinetic (PK) parameter estimates (points) for each monoclonal antibody pair of parental vs. LS by PK parameter. The following PK parameters (top to bottom) are displayed: clearance (CL) (L/day), central volume (Vc) (L), inter-compartmental clearance (Q) (L/day), and peripheral volume (Vp) (L). Colors of the points indicate different administration routes. In all plots, box plots indicate median, interquartile range (IQR) (box), and 1.5× IQR (whiskers). The median value is listed above each box plot.

**Figure 2 pharmaceutics-16-00594-f002:**
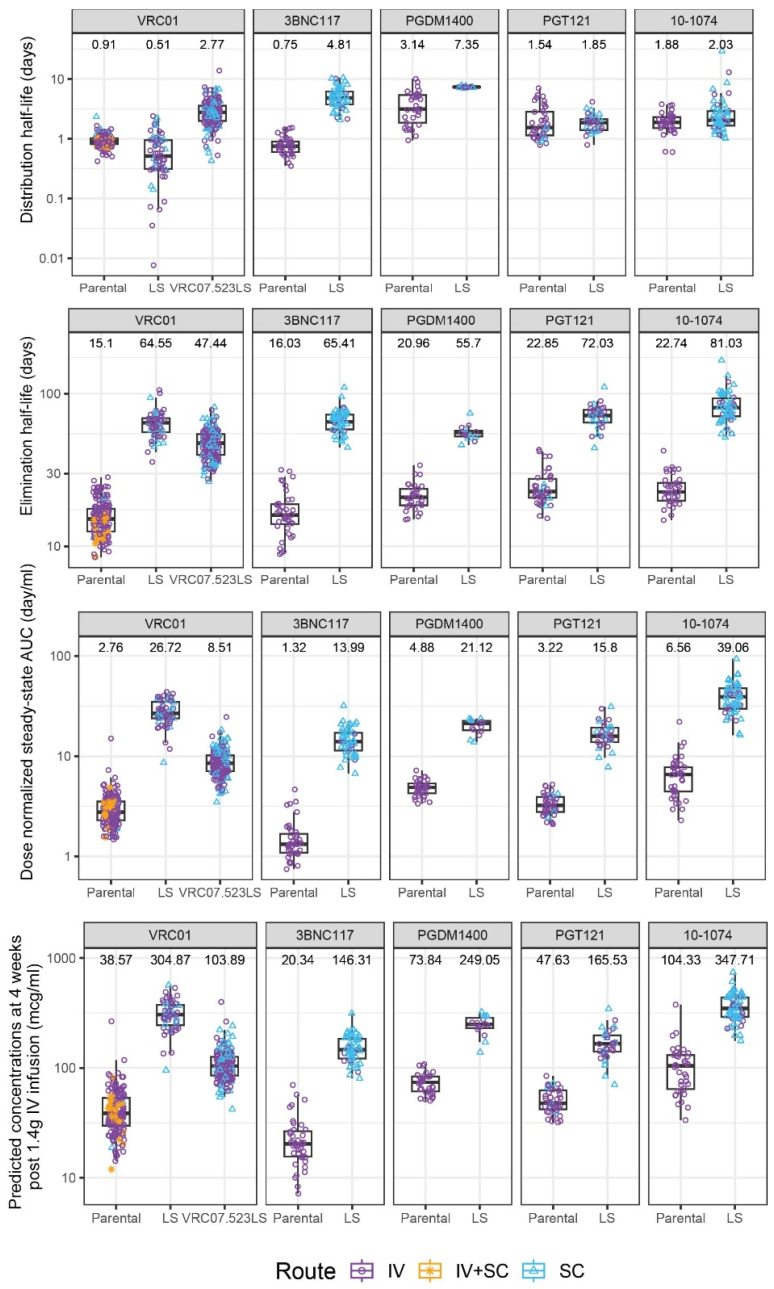
Distributions of population-PK-model-estimated covariate-unadjusted individual-level PK features for each monoclonal antibody pair of parental vs. LS. The following PK features (top to bottom) are displayed: distribution half-life (days), elimination half-life (days), dose-normalized steady-state area under the curve (AUC) (day/mL), and estimated concentrations at 4 weeks post-infusion (mcg/mL). Colors of the points indicate different administration routes. In all plots, box plots indicate median, interquartile range (IQR) (box), and 1.5× IQR (whiskers). The median value is listed above each box plot.

**Figure 3 pharmaceutics-16-00594-f003:**
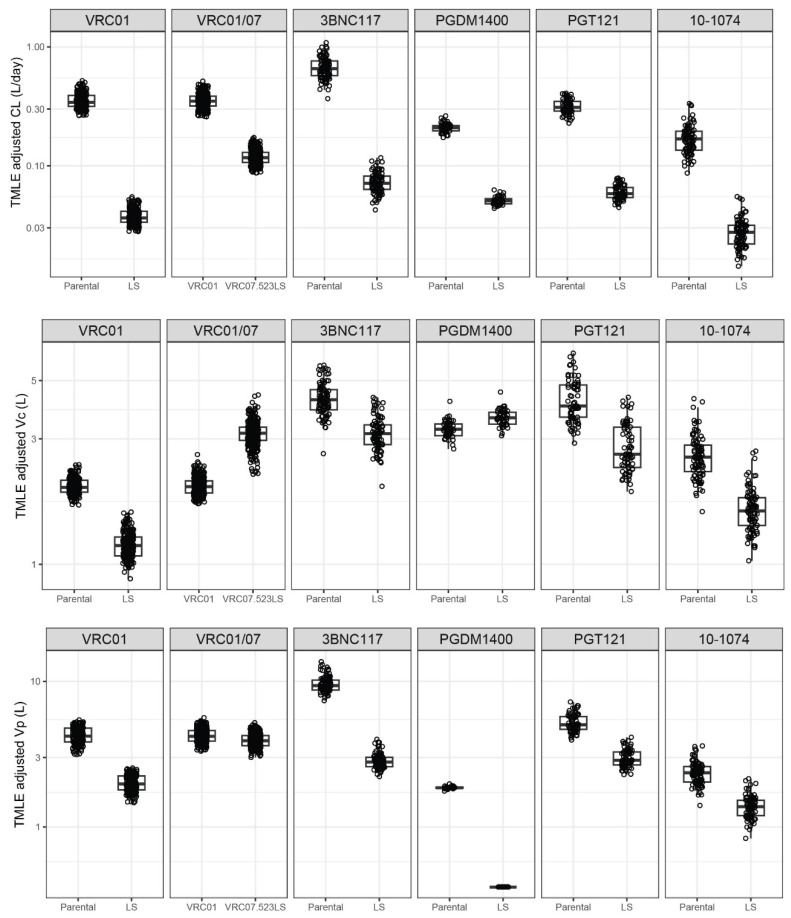
Distributions of targeted maximum likelihood estimation (TMLE)-adjusted individual-level pharmacokinetic (PK) parameter estimates (points) for each monoclonal antibody pair of parental vs. LS by PK parameter. The following PK parameters (**top** to **bottom**) are displayed: clearance (CL) (L/day), central volume (Vc) (L), peripheral volume (Vp) (L). In all plots, box plots indicate median, interquartile range (IQR) (box), and 1.5× IQR (whiskers). Each dot indicates the TMLE-adjusted PK feature estimate for each participant as if all received the parental mAb (left) or all LS mAb (right) under a causal framework.

**Figure 4 pharmaceutics-16-00594-f004:**
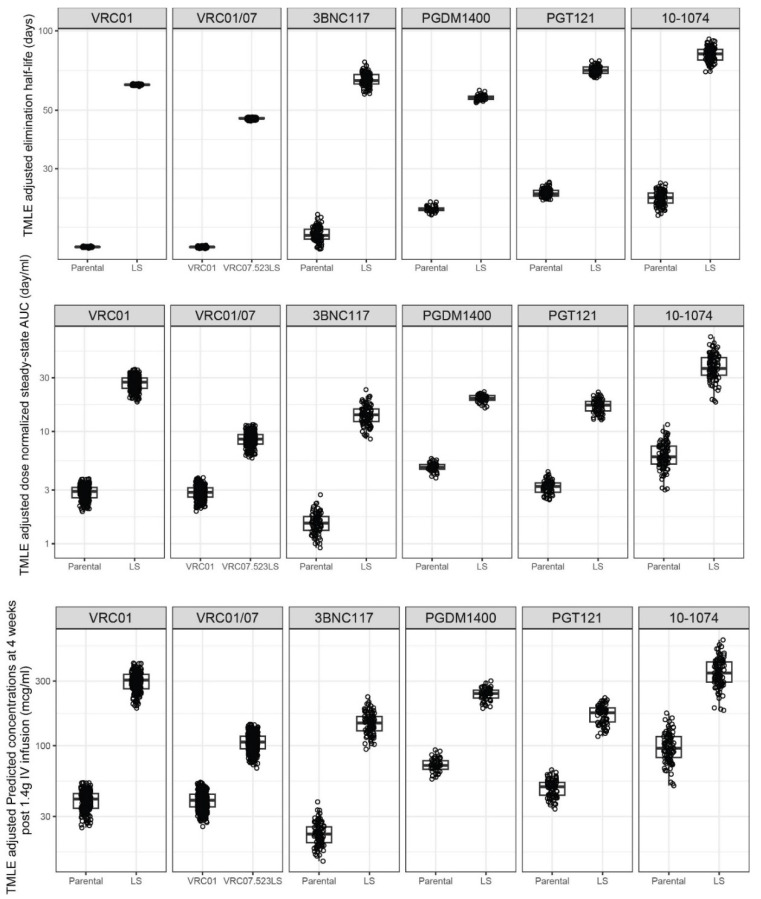
Distributions of targeted maximum likelihood estimation (TMLE)-adjusted individual-level PK features for each mAb pair of parental vs. LS by PK feature. The following PK features (**top** to **bottom**) are displayed: distribution half-life (days), elimination half-life (days), dose-normalized steady-state area under the curve (AUC) (day/mL), and estimated concentrations at 4 weeks post-infusion (mcg/mL). In all plots, box plots indicate median, interquartile range (IQR) (box), and 1.5× IQR (whiskers). Each dot indicates the TMLE-adjusted PK feature estimate for each participant as if all received the parental mAb (left) or all LS mAb (right) under a causal framework.

**Figure 5 pharmaceutics-16-00594-f005:**
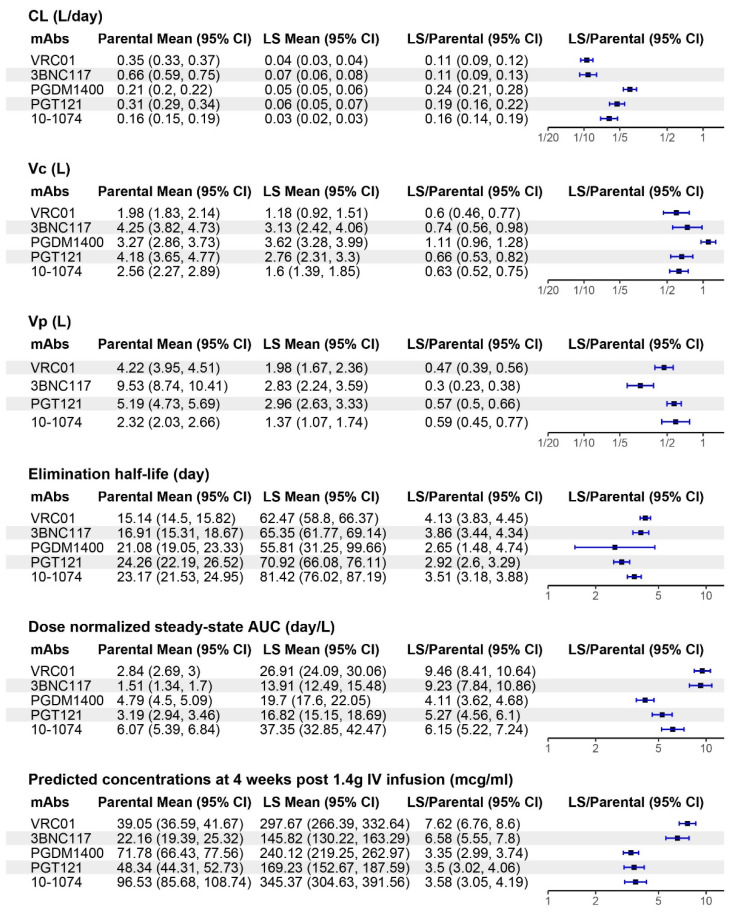
Mean and fold-change estimates between parental and LS variants for each pharmacokinetic feature. For each PK feature, the means for the parental and for the LS variant, LS versus parental (LS/parental) fold change and the associated 95% CIs were estimated using the target maximum likelihood estimation (TMLE) method. The fold-change was computed by exponentiating the average treatment effect estimated on log-normally distributed PK features. In the last column, the points indicate the estimated fold-change of the given parameter comparing the LS to the parental variant for the given monoclonal antibody with the error bars indicating the 95% CI obtained via a bootstrap procedure as described in the Materials and Methods section.

**Figure 6 pharmaceutics-16-00594-f006:**
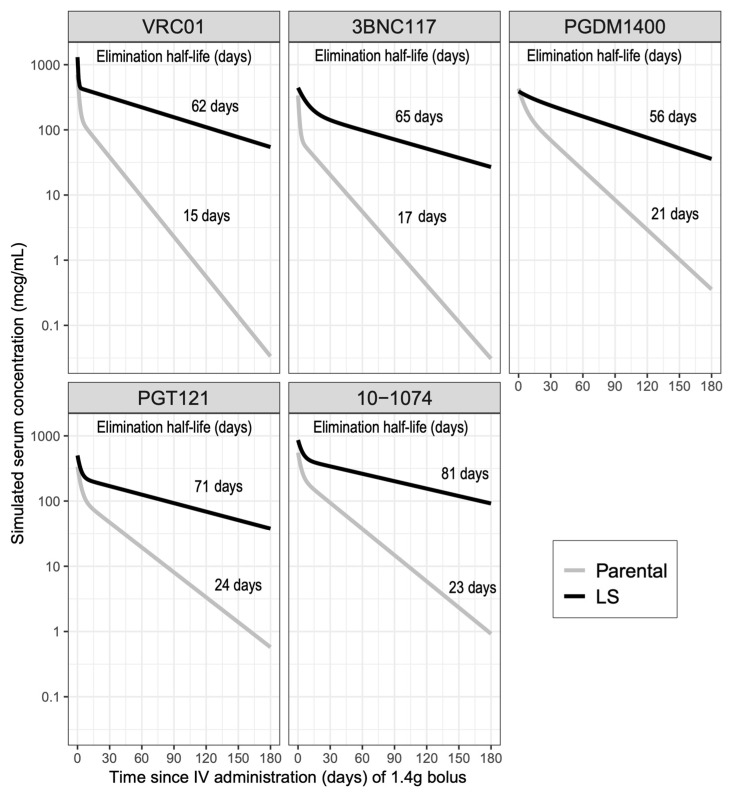
Simulated serum concentrations (mcg/mL) for each monoclonal antibody parental and LS variant pair. Gray curves are for parental mAbs and black curves for LS variants. Values shown on top of each curve are the estimated mean elimination half-life (days) for the corresponding mAb. The mean pharmacokinetic (PK) parameters estimated via targeted maximum likelihood estimation (TMLE) were used in the simulations, assuming a 1.4 g dose administered via bolus intravenous infusion. When a PK parameter was not estimated with TMLE (inter-compartmental clearance (Q) for all antibodies and peripheral volume (Vp) for PGDM1400LS), the mean parameter estimates from the covariate-unadjusted model were used.

**Table 1 pharmaceutics-16-00594-t001:** Description of the 16 phase 1 clinical trials included in the study. Note: only data from intravenous and subcutaneous delivery routes were included in the analyses. mAb, monoclonal antibody; IV, intravenous; SC, subcutaneous; PK, pharmacokinetics.

mAb	HIV-1 Epitope Specificity	Clinical Trial (clinicaltrial.gov Registration #)	Study Country(ies)	Single or Combination mAb Regimens	IV Administration (n)	SC Administration (n)	PK Timepoints after Administration through the First 4 Weeks of the Study **	Assay	References
VRC01	CD4 binding site	VRC602 (#NCT01993706)	USA	Single	18	5	End of infusion, 1, 2, 4, 8, 12, 24, and 72 h, and 2, 7, 14, 21, and 28 days	ELISA (VRC)	[25]
		HVTN 104 (#NCT02165267)	USA	Single	64	20	1 h, 3 days, and 2 and 4 weeks	ELISA (VRC)	[26]
		HVTN 116 (#NCT02797171)	USA/RSA	Single	62	0	Weeks 1-2	Singulex (VRC)	[30]
VRC01LS		VRC606 (#NCT02599896)	USA	Single	21	18	1, 2, 6, 24, 48, and 72 h, and 1, 2, 3, and 4 weeks	ELISA (VRC)	[19]
		HVTN 116 (#NCT02797171)	USA/RSA	Single	17	0	Weeks 1-2	Singulex (VRC)	[30]
VRC07-523LS		VRC605 (#NCT03015181)	USA	Single	17	8	End of infusions, 1, 2, 6, 24, 48, and 72 h, and 1, 2, 3, and 4 weeks	ELISA (VRC)	[20]
		HVTN 127/HPTN 087 (#NCT03387150)	USA/SUI	Single	59	41	3, 6, and 28 days	ELISA (VRC)/ BAMA (Duke)	[31,32]
		HVTN 130/HPTN 089 (#NCT03928821)	USA	Combination	26	0	3, 6, 14, and 28 days	BAMA (Duke)	[28]
		HVTN 136/HPTN 092 (#NCT04212091)	USA	Combination	10	10	1 h, and 1, 2, 3, 6, 14, and 28 days	BAMA (Duke)	[33]
3BNC117		MCA-0835/Nussenzweig 446 * (#NCT02018510)	USA	Single	22	0	End of infusion, 0. 5, 3 6, 9, and 12 h; and 1, 2, 4, 7, 14, 21, and 28 days	ELISA (Celldex)	[34]
		YCO-0899/Nussenzweig 583 (#NCT02824536)	USA	Combination	18	0	End of infusion, 1, 2, 7, 14, and 28 days	ELISA (Duke, transferred from Celldex)	[23]
3BNC117-LS		YCO-0946/Nussenzweig 684 * (#NCT03254277)	USA	Single	3	12	End of infusion/injection, 1 (SC), 3, 7, 14, and 28 days	ELISA (Duke, transferred from Celldex)	[27]
		YCO-0971/Nussenzweig 739 * (#NCT03554408)	USA	Combination	5	30	End of infusion/injection, 1, 3, 7, 14, and 28 days	ELISA (Duke, transferred from Celldex)	[35]
PGDM1400	V1V2 Glycan	IAVI T002/Barouch 693 * (#NCT03205917)	USA	Single	9	0	End of infusion, 3, 6, and 24 h, and 1, 2, 3, 7, and 12 days	BAMA (Duke)	[24]
		IAVI T002/Barouch 693 * (#NCT03205917)	USA	Combination	12	0	End of infusion, 3, 6, and 24 h, and 1, 2, 3, 7, and 12 days	BAMA (Duke)	[24]
		HVTN 130/HPTN 089 (#NCT03928821)	USA	Combination	15	0	3, 6, 14, and 28 days	BAMA (Duke)	[28]
PGDM1400LS		HVTN 140/HPTN101 (Part A) (#NCT05184452)	USA/KEN/ RSA/ZIM	Single	9	6	1 h, and 3, 6, and 28 days	BAMA (Duke)	[36]
PGT121	V3 Glycan	IAVI T001/Barouch 628 * (#NCT02960581)	USA	Single	12	4	End of infusion, 30 min, 3, 6, 9, and 12 h, and 1, 2, 3, 7, 14, 21, and 28 days	BAMA (Duke)	[29]
		IAVI T002/Barouch 693 * (#NCT03205917)	USA	Combination	12	0	End of infusion, 3, 6, and 24 h, and 1, 2, 3, 7, and 12 days	BAMA (Duke)	[24]
		HVTN 130/HPTN 089 (#NCT03928821)	USA	Combination	15	0	3, 6, 14, and 28 days	BAMA (Duke)	[28]
PGT121.414.LS		HVTN 136/HPTN 092 (# NCT04212091)	USA	Single and combination	20	13	0, 1, 2, 3, 6, 14, and 28 days	BAMA (Duke)	[33]
10-1074		MCA-0885/Nussenzweig 514 * (#NCT02511990)	USA	Single	14	0	End of infusion, 30 min, 3, 6, 9, 12 h, and 1, 2, 4, 7, 14, 21, and 28 days	ELISA (Celldex)	[37]
		YCO-0899/Nussenzweig 583 (#NCT02824536)	USA	Combination	18	0	End of infusion, 2 days, 1, 2, and 4 weeks	ELISA (Duke, transferred from Celldex)	[23]
10-1074-LS		YCO-0971/Nussenzweig 739 * (#NCT03554408)	USA	Single	9	12	End of infusion/injection, 1, 3, 7, 14, and 28 days	ELISA (Duke, transferred from Celldex)	[35]
		YCO-0971/Nussenzweig 739 * (#NCT03554408)	USA	Combination	5	30	End of infusion/injection, 1, 3, 7, 14, and 28 days	ELISA (Duke, transferred from Celldex)	[35]

* Only PK data from groups without HIV were included in all presented analyses. ** All available timepoints were used for analysis; for brevity only time points collected in the first 4 weeks were listed here. For more information, please review documentation for each study publicly available, or by request. All studies complied with relevant ethical requirements including obtaining participant consent and were registered with ClinicalTrials.gov.

## Data Availability

Please refer to the study-specific publications for information on which data are available for each of the 16 clinical trials. The combined dataset can be made available upon request.

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
