# Peer review of "Impact of LS Mutation on Pharmacokinetics of Preventive HIV Broadly Neutralizing Monoclonal Antibodies: A Cross-Protocol Analysis of 16 Clinical Trials in People without HIV"

_pharmaceutics, 2024, doi:10.3390/pharmaceutics16050594_

Round 1
Reviewer 1 Report (Previous Reviewer 1)
Comments and Suggestions for Authors
I have no issues.
Reviewer 2 Report (Previous Reviewer 2)
Comments and Suggestions for Authors
The authors have corrected the issues that were noticed during the previous submission of the article. In its present form, the reviewer sees no reason not to recommend the article for publication
This manuscript is a resubmission of an earlier submission. The following is a list of the peer review reports and author responses from that submission.
Round 1
Reviewer 1 Report
Comments and Suggestions for Authors
Bryan T. Mayer conducted a study to investigate whether the "LS" mutation in the Fc region of anti-HIV IgG1 mAbs (VRC01/LS/VRC07-523LS, 3BNC117/LS, PGDM1400/LS, PGT121/LS, 10-1074/LS) could enhance the pharmacokinetic (PK) profiles of mAbs. The data from 16 clinical trials involving 583 participants without HIV were analyzed, revealing that LS mAbs improved several PK parameters, such as increases in elimination half-life, dose-normalized area under the curve, and predicted concentration. Furthermore, the findings suggest a favorable PK profile of LS variants irrespective of HIV epitope specificity. This study advocates for lower dosages and/or less frequent dosing of LS variants to achieve comparable levels of antibody exposure for future anti-HIV LS mAbs. However, I have concerns about the originality of this research as similar mutation modifications have been applied to various antibodies, including those targeting HIV and malaria.
1 My primary criticism is that the author focused solely on HIV antibodies and did not include data on other antibodies.
2 While the study employed various analytical methods to validate the conclusions, I perceive this work more as a review than an original research article, raising concerns about its originality due to similar mutation applications in different antibodies, such as those targeting HIV and malaria.
Reviewer 2 Report
Comments and Suggestions for Authors
An article submitted by Bryan T. Mayer, Yunda Huang, and other co-authors in Pharmaceutics analyzes the production of bnAbs in patients from 16 clinical trials of HIV treatment.
First of all, the reviewer wants to ask the authors what hypothesis and assumption they tested in this work.
Second, it is not clear from the text of the article how patients were included in the study, what manipulations were performed on them, whether permission from bioethical committees was obtained, etc.
The reviewer also had several comments and questions, the answers to which should be given in the text of the article (not only in the form of a response to the reviewer):
1) Abbreviations should not be used in the Abstract
2) LS mutations should be explained in the Abstract, at least in one sentence
3) line 50 - "as of April 2021", this fact should be updated as of February 2024
4) line 70 - “constant fragment crystallizable” one of the epithets should be left in the article
5) The data presented in Table 1 and Table 2 can easily be transferred (Table 2) to the Supplementary. It is unlikely that a reader will be interested in this information, which is secondary to the manuscript content. Moreover, the authors do not refer to the results presented in these tables within the text of the manuscript (except for the single mentions of Table 1 in line 244, and Table 2 in line 252)
6) Was a statistical analysis of the significance of the differences in the data presented in Figures 1 and 2 carried out? If not, then what is the point of presenting these results?
7) The data presented in Table 3 are extremely poor; this table should be reformatted, leaving the “Parameter” rows (the same for all columns) and adding corresponding mAb columns. In its current form, Table 3 does not deserve to be left in the manuscript since even an attentive reader cannot derive any benefit from it.
8) It is also unclear from the text of the article what information presented in Fig. 5 deserves the reader's attention. A similar collection of figures belongs in the Supplementary.
The authors who wrote the manuscript probably had no questions about the presentation. But in this form, the article is unlikely to interest readers. The form and meaning of the content should be thoroughly reworked.
Reviewer 3 Report
Comments and Suggestions for Authors
To illustrate the LS mutation for mAbs targeting HIV, authors examined pharmacokinetic parameters between parental versus LS-variants for 5 pairs of anti-HIV IgG1 mAbs with data from 16 clinical trial of 583 participants without HIV. Results demonstrated that lower clearance rate, central volume, and peripheral volume of distribution for all LS variants compared to parental mAbs. These results support lower dosages or less frequent dosing of LS variants to achieve similar levels of antibody exposure for future anti-HIV LS mAbs. It is so interesting for reader. These help the application of monoclonal antibodies for LS mutations.
Major comments:
1 the abstract should be rewritten including main method, results, conclusions.
2 the discussion is not enough, and it should be deeply performed according to the main findings.
Minor comments:
Line 28, the abbreviation first time appears, which should use whole name.
Line 181, LLOQ
Comments on the Quality of English LanguageModerate editing of English language required